# Neurological Disorders and Clinical Progression in Boxers from the 20th Century: A Narrative Review [note 1]

**DOI:** 10.3390/brainsci15070729

**Published:** 2025-07-08

**Authors:** Rudolph J. Castellani, Nicolas Kostelecky, Jared T. Ahrendsen, Malik Nassan, Pouya Jamshidi, Grant L. Iverson

**Affiliations:** 1Department of Pathology, Northwestern University Feinberg School of Medicine, Chicago, IL 60611, USA; jared.ahrendsen@nm.org (J.T.A.); pouya.jamshidi@nm.org (P.J.); 2Department of Pathology, Western Michigan University Homer Stryker M.D. School of Medicine, Kalamazoo, MI 49007, USA; nicolas.kostecky@wmed.edu; 3Mesulam Center for Cognitive Neurology and Alzheimer’s Disease, Departments of Neurology and Psychiatry and Behavioral Sciences, Northwestern University Feinberg School of Medicine, Chicago, IL 60611, USA; malik.nassan@northwestern.edu; 4Department of Physical Medicine and Rehabilitation, Harvard Medical School, Boston, MA 02115, USA; giverson@mgh.harvard.edu; 5Department of Physical Medicine and Rehabilitation, Spaulding Rehabilitation Hospital, Charlestown, MA 02129, USA; 6Department of Physical Medicine and Rehabilitation, Schoen Adams Research Institute at Spaulding Rehabilitation, Charlestown, MA 02129, USA; 7Home Base, A Red Sox Foundation and Massachusetts General Hospital Program, Charlestown, MA 02129, USA; 8Mass General for Children Sports Concussion Program, Boston, MA 02114, USA

**Keywords:** traumatic brain injury, punch drunk syndrome, dementia pugilistica, traumatic encephalopathy syndrome, chronic traumatic encephalopathy

## Abstract

**Introduction**: There are no validated clinical diagnostic criteria for chronic traumatic encephalopathy or traumatic encephalopathy syndrome (TES). To understand the historical clinical condition, its applicability to modern day athletes, and the pathogenesis of clinical problems, we examined the literature describing boxers from the 20th century, with specific attention paid to neurological findings and characteristics of clinical disease progression. **Methods**: Data were extracted for 243 boxers included in 45 articles published between 1928 and 1999, including cases from articles originally published in German. The presence or absence of 22 neurological signs and features were extracted. **Results**: The most common neurological problems were slurring dysarthria (49%), gait disturbances (44%), and memory loss (36%), with several other problems that were less frequent, including hyperreflexia (25%), ataxia (22%), increased tone (19%), and extensor Babinski sign (16%). Frank dementia appeared in some cases (17%). There were significantly fewer neurological deficits reported in boxers who fought in the latter part of the 20th century compared to boxers who fought earlier in the century. For more than half of the cases, there were no comments about whether the neurological problems were progressive (145, 60%). A progressive condition was described in 71 cases (29%) and a stationary or improving condition was described in 27 cases (11%). Canonical neurodegenerative disease-like progression was described in 15 cases (6%). **Discussion**: Neurological problems associated with boxing-related neurotrauma during the 20th century are the foundation for present-day TES. However, the clinical signs and features in the 20th century differ in most ways from the modern criteria for TES.

## 1. Introduction

Neurological problems from boxing-related neurotrauma were first described in the medical literature in 1928, in Martland’s seminal article on early 20th-century boxers [1]. In this article, Martland documented the clinical condition of twenty-three boxers with the assistance of a fight promoter (see the table on page 1106 of Martland, 1928) [1]. Fifteen men were labeled ”punch drunk”. Four were listed as having various neurological problems (e.g., ”Parkinsonian syndrome”, ”drags leg”, ”bad shape”, ”talks slow”, and ”thinks slow”), and four were simply listed as ”asylum”. Over the next 70 years, many additional articles and reviews were published that described neurological problems in boxers (see Table 1).

The various accounts of punch drunk syndrome suggested a peculiar and heterogeneous array of neurological signs and problems. Motor signs appeared to be particularly common and were often recorded while boxers were still competing [32]. In severe forms, frank dementia occurred, sometimes with loss of insight into the neurological deficits [5,23,32] and sometimes necessitating lifelong commitment to a long-term care facility [1,34]. In one unfortunate biographical account, a boxer around the age of 30 continued to have promoted fights and continued to compete *after* being institutionalized for dementia [47]. He remained a ward of the state until his death at age 67.

The peculiarity of the condition and the finding of progressive symptomatology in some cases led to speculation that boxing-related neurotrauma might underlie, or be related to, a neurodegenerative disease. In 1949, Critchley noted the following with respect to progression: “This is of great interest and importance and rather suggests a slowly progressive pathological process which has been fired off by the original injuries” (page 134) [9]. Descriptions of autopsy neuropathology in boxers, however, were sparse, with the largest case series, published in the 1970s (Corsellis et al.), consisting of only 15 cases [34]. Sporadic neurodegenerative diseases (e.g., Alzheimer’s disease, Lewy body dementia, progressive supranuclear palsy) were present among those cases according to a recent re-analysis of the Corsellis series [48] (see Appendix A).

Modern studies on chronic traumatic encephalopathy (CTE) and traumatic encephalopathy syndrome (TES), published over the past 15 years, often refer to the 20th-century literature on ultra-high exposure boxers as foundational for understanding clinical problems experienced by current and former athletes; thus, an understanding of the classical literature on these boxers is important for understanding modern criteria for TES. Neurotrauma in these boxers was extreme, particularly before World War II, and resulted in a florid clinical syndrome, obvious even to lay observers, characterized by slurring dysarthria, gait disturbances, increased reflexes, and memory loss, among other neurological signs.

The issue of disease progression, however, was complex. Early articles by Martland (1928) [1], Parker (1934) [2], Carroll (1936) [49], Bowman and Blau (1940) [5], and Critchley (1949) [9] all alluded to stationary disease in some cases, whereas symptomatic improvement was also described. Roberts [32], who examined the issue of progressive disease in the largest study on boxers to date, also noted that most cases remained stationary after retirement, and there were examples of “undoubted improvement.” This is notable because canonical neurodegenerative diseases are invariably progressive. Given the complexity of punch drunk syndrome, traumatic encephalopathy, and dementia pugilistica as described last century, and how the clinical condition of those boxers served as the foundation for modern day TES, we wanted to examine neurological signs and disease progression in detail. TES criteria incorporate, for example, delayed-onset deterioration (i.e., a latent period) and a progressive course [50]. If, as Roberts suggested in 1969, most boxers were symptomatic while still competing, and that the course was not progressive in most cases, the current TES conceptualization would differ from what was experienced by the highest exposure boxers last century.

The objective of this narrative review is therefore to provide a detailed account of the diverse neurological problems, the relative frequency of those problems, and the characteristics of disease progression in boxers from an era of high neurotrauma exposure. In doing so, we hope to provide insight into the condition upon which current concepts of chronic traumatic encephalopathy (CTE) and traumatic encephalopathy syndrome (TES) are based. We also address whether there was a change in the frequency of reported neurological signs over time in the 20th century. Other authors have documented a substantial reduction in boxing exposure from the early to the late 20th century [51]. We chose this period because neurotrauma exposure in boxers was extreme, and the clinical manifestations were more apparent early in the 20th century, which dissipated over time with decreasing exposure. Therefore, in addition to enumerating the neurological signs in boxers, we examined whether there appeared to be a so-called “dose response” related to boxing exposure and neurological deficits.

## 2. Literature Review and Data Extraction

We relied on a narrative review methodology, not a systematic review, for this article. Data were extracted from 45 articles [1,2,3,4,5,6,7,8,9,10,11,12,13,14,15,16,17,18,19,20,21,22,23,24,25,26,27,28,29,30,31,32,33,34,35,36,37,38,39,40,41,42,43,44,45] including 243 boxers (see Table 1 and see Appendix A). All articles that (i) described one or more boxers and (ii) had some accompanying neurological data (even if normal) were included. No article or boxer was excluded based on presumed neurological diagnosis (traumatic encephalopathy, paralysis agitans, neurosyphilis, etc.), because the goal was to include all neurological information in the boxer literature. It is inevitable that some cases are misdiagnoses, but to capture the depth and breadth of available information, we erred on the side of inclusion rather than exclusion.

Data from 30 boxers included in fourteen of these 45 articles were obtained from partial translations in a book (*Boxing: Medical Aspects* [46] published in 2003 [3,4,6,7,11,13,14,15,16,17,19,21,25,31]) because the full length article was not obtainable, and these cases were not included in prior reviews of the literature [52,53,54,55]. The articles by Constantinidis and Tissot (1967) [28] and Grahmann and Ule (1957) [17] were translated to English using Google Translate. Of the 243 boxers, the articles indicated professional status in 166 boxers and amateur status in 36 boxers. Six were said to have engaged in sparring.

Data extracted from four case histories in Critchley’s 1949 article (case C, D, E, and F) [9] appeared to have been duplicated in his 1957 article [18] and were extracted only once using the 1949 article. The narratives in the 1949 versus the 1957 articles differed slightly in these four cases, but the differences did not impact the coding. The boxer reported by Constantinidis and Tissot in 1967 [28] was also reported by Hof et al. in 1992 [40]. Data were extracted from the 1967 article, which was more detailed. We extracted only four of 15 cases from Johnson (1969), cases 1, 6, 10, and 17 [33]. These four case descriptions contained neurological findings, whereas the other cases were not described individually. In addition, there was an overlap between cases described in Johnson’s 1969 article and those described in Mawdsley and Ferguson (1963) [24]. One of the cases included in our data extraction from Johnson (case 1) might overlap with one of the cases described in Mawdsley and Ferguson (case 1).

Our research group recently published a narrative review of psychiatric features of traumatic encephalopathy syndrome as conceptualized in the 20th century [54] and a review that applied the new consensus criteria for traumatic encephalopathy syndrome [50] to cases of boxers from the 20th century [55]. Those two articles also focused on 20th-century boxers and included 21 published studies and 155–157 case descriptions. The current narrative review added 88 additional cases. As such, we added articles that were not published in English, including articles translated from *Boxing: Medical Aspects* and others [3,4,5,6,7,8,11,12,13,14,15,16,17,19,21,25,26,27,28,30,31]. In addition, we included more cases from Roberts (1969) [32] and more cases from Critchley (1957) [18]. With respect to Roberts, in his book examining a random sample of 224 former boxers in the United Kingdom, he included detailed case histories in 11 subjects he believed to be symptomatic for traumatic encephalopathy, which we included in this review. Roberts also included limited details from groups of cases with similar findings, which we included as well. For example, with respect to “Case 7 (sample no. 63)” (as designated by Roberts), a quote is provided below:

“There were seven others (Sample Nos. 4, 14, 29, 67, 133, 169, 216) whose ages ranged from 38 to 58 who presented a similar clinical picture of dysarthria and disequilibrium with asymmetrical ataxia as evidence of cerebellar lesions, together with pyramidal lesions and immobile facies, but with little or no overt intellectual defect” (page 37) [32].

Given the neurological details provided, these additional seven cases were included as individual cases for the data extraction. Likewise, Roberts’ descriptions of Cases 8, 9, 10, and 11 each include a paragraph describing additional cases with a similar clinical picture, which we included for data extraction. Roberts also included details of 11 cases that he diagnosed as having other neurological diseases (not traumatic encephalopathy). Given the boxing exposure, the details of the examination, and the possibility that boxing-related neurological deficits co-existed with other conditions, these cases were included in the data extraction as well. In total, data were extracted from 48 cases from the book by Roberts.

Data relating to neurological signs were analyzed from the case histories (Table 2). If the specific data point was not mentioned one way or the other, it was coded as not mentioned “NM” (see Appendix A). Each item was coded as binary (present or absent), with some extractions further itemized according to the footnotes in Appendix A. If a specific data point was not mentioned, then it was considered absent for the statistical analyses.

Inferences were avoided. For example, if a clinical vignette stated that the “physical examination was normal” but the specific neurological signs or symptoms were not mentioned, it was coded as “not mentioned.” If the vignette indicated that the specific neurological sign could be inferred from a specific normal finding, it was coded as absent. For example, if a vignette stated that “speech was normal”, then dysarthria was coded as absent. All coding was conducted initially by RJC, who reviewed each article and coded the neurological problems (Table 2). This initial coding was reviewed by NK for disagreements or oversights. There was a total of 15 disagreements. After resolving the disagreements, the coding was reviewed by two additional authors (JTA, PJ) who did not identify additional areas of disagreement in the data extraction.

## 3. Neurological Problems in 20th-Century Boxers

Neurological problems reported most frequently in 20th-century boxers were dysarthria, gait disturbances, and memory loss (see Table 2). Other neurological problems were somewhat less commonly reported. These included hyperreflexia, the Babinski sign (plantar extensor reflex), increased tone or spasticity, abnormal facies, tremor, ataxia, eye movement disorders, bradykinesia or slowness of movements, and headache. Hyperreflexia was asymmetric in 25 out of 61 cases (41%), of which 17 were left greater than right and 8 were right greater than left. Roberts commented on this phenomenon and the possibility that this tendency for asymmetry might relate to boxing-related neurotrauma (page 85) [32].

Abnormal facies were typically described as immobile, impassive, or mask-like. Tremor was not further qualified in some articles and was described as intention tremor specifically in some articles. Tremor usually referred to the upper extremities (arm and/or hand), although sometimes involved head and neck [20], tongue [1], and lower extremities [13]. Other signs that were less commonly reported include vision loss, positive Romberg sign, cogwheel rigidity, dysdiadochokinesis, seizures, clonus, and pathological drooling or dribbling.

## 4. Disease Progression in 20th-Century Boxers

Data relating to disease progression were extracted from the case histories (Table 3 and Figure 1). Specific quotations from the articles are provided in Appendix A. Data extraction and case ratings were performed initially by RJC and NK, followed by discussion of discrepancies and agreement on ratings. The ratings were then reviewed for agreement by four authors (RJC, NK, PJ, JTA). 

The descriptions of progression varied from crude, to detailed, to no mention of progression. In some individual cases, a period of follow up was provided such that more than one examination was considered in the assessment [2,5,6,7,10,12,17,20,28,33]. The nature of the progression was variable. We divided progression into multiple categories as follows: (i) progression not mentioned; (ii) no progression; (iii) symptomatic improvement; (iv) progressive symptomatology limited to motor function or memory function; (v) complex progression over decades with medical co-morbidities such as alcoholism, vascular disease, and traumatic brain injury sustained outside of boxing; and (vi) progression over a period of years beginning after retirement, with apparently preserved neurological function at baseline (i.e., progression similar to known neurodegenerative diseases).

The issue of progressive disease was discussed in 88 out of the 243 boxers (36%). The studies were not cross-sectional prevalence studies (with the exception of Roberts [32]), case-control studies, or cohort studies; thus, the incidence or prevalence of a progressive condition or disease among 20th-century boxers is unknown. Our data are limited to the frequency with which such characteristics appeared in the articles.

Because the issue of progressive disease was not addressed in the majority of articles, and because many articles did not state specifically whether the boxer was amateur or professional, we did not analyze professional versus amateur status with respect to disease progression.

### 4.1. Stationary Condition or Symptomatic Improvement

Progressive disease was rated as something other than ”not mentioned” 98 times in 88 boxers (10 boxers had more than one rating) (Table 3). When discussing percentages, both 88 and 243 are used as the denominator in Table 3. Thus, articles described absence of progression in 19/88 boxers (22%, 8% of the total sample) and symptomatic improvement in 8/88 boxers (9%, 3% of the total sample). There were 71/88 boxers (81%, 29% of the total sample) who were described as showing evidence of some type of progression.

### 4.2. Progression Limited to Motor Function, With or Without Memory Loss

There were 21 boxers (24% of those with progression, 9% of the total sample) who were rated as having progressive disease limited to motor signs, such as tremor, gait dysfunction, and dysarthria. Some examples include the following: (i) Mawdsley and Ferguson, Case 2—“unsteadiness of his gait slowly worsened”; (ii) Mawdsley and Ferguson, Case 5—“When he was about 35 his gait became unsteady and his speech became slurred. The symptoms have progressed”; (iii) Roberts, Case 3—“Both he and his wife have been aware that his speech, for at least the past ten years and perhaps longer, had been ’thick and getting thicker’”; and (iv) Roberts, Case 9—“He himself was not aware of this but had noticed that he was unsteady on his feet during his last year’s boxing and that this had become worse in recent years so that he often lost his balance”.

Also of note are cases in which motor signs progressed and then stabilized. For example, Critchley noted in his case 8 ”Three months later his hands became tremulous. His symptoms increased up to a point and then remained stationary”. Three cases reported by Parker described neurological deficits limited to motor function, each showing some degree of symptomatic improvement [2]. Motor disturbances were combined with memory loss in seven boxers (8% of those with progression, 3% of the total sample). An example is a case described by Johnson in 1969, with a quote reprinted below:

“A 60-year-old light-weight, ex-professional boxer retired at 30 years from the ring when he developed severe intention tremor of the right hand, ataxic gait and poor memory for recent events. The condition progressed for five years but has since been relatively stationary” (page 48) [33].

### 4.3. Isolated Progression of Memory Loss

Progressive change limited to memory function was noted in only four boxers in the 20th-century literature (5% of those with progression, 2% of the total sample). Case 3 in Mawdsley and Ferguson, for example, was described as follows: “At the age of 50 he and his family noticed a deterioration in his memory. This defect was severe for recent events and progressed” (page 796) [24]. Another example of isolated progression of memory dysfunction was described by Roberts in 1969, reprinted below:

“He and his wife thought his memory has been poor since the end of his boxing career and had progressively worsened over the years, so that, for many years, he had to rely entirely on his wife, forgetting, himself, almost anything he was told immediately. Neither he nor his wife were aware of any other neurological disabilities” (page 122) [32].

Deterioration in memory is clear from the narrative, as is the absence of any other neurological signs. Isolated progressive memory impairment, however, was uncommon. Roberts made a similar observation and included this case as an example of memory loss in a boxer, possibly unrelated to boxing, reprinted below:

“This was the only individual in the random sample with clear evidence of a severe memory defect who did not also have evidence of extensive lesions elsewhere in his central nervous system. Therefore, although it seems likely that his forgetfulness first became apparent at the time he gave up boxing and was possibly related to this career, in the absence of any of the other characteristic features of the syndrome of traumatic encephalopathy exhibited by the remainder of the series it has not been assumed unquestioningly that his amnestic syndrome was due to boxing” (pages 122–123) [32].

### 4.4. Complex Progression

There were 24 boxers (27% of those with progression, 10% of the total sample) who were coded as having complex progression. In these cases, there was clear evidence of deterioration, but it was often associated with substantial neurological deficits at or prior to retirement from boxing. The narratives often indicated progression over several decades, and there were often co-morbidities such as chronic alcoholism, vascular disease, difficulty functioning in society (e.g., homelessness, tendency toward violence), and traumatic brain injury sustained outside the ring. Eight of the 15 boxers in the Corsellis et al. series [34] were coded as having complex progression. Case 1 is illustrative of this type of case, with a quote reprinted below:

“He married in his early 20s when he already become a social as well as a boxing success. Soon his life became more hectic and ‘he changed completely’. He wenched and drank and gambled heavily. His memory began to fail him. He had three car accidents; in one he suffered severe scalp lacerations and an injury to the right eye, followed by three months in the hospital. His marriage broke up and he drifted away from his family, only returning for an occasional embarrassing visit. He had violent outbursts, he was ‘knocked out’ by only a small amount of alcohol, his behaviour was ‘disgusting’. His brother remarked that ‘his brain was not functioning—he made mistakes in reckoning’. He could not settle in a job and he became a vagrant. In his 50s his jaw was broken in a brawl. At the age of 62 he was found lying neglected and louse-ridden in the boiler house of a hotel” (page 271) [34].

### 4.5. Neurodegenerative Disease-like Progression

There were 15 boxers (17% of those with progression, 6% of the total sample) who were rated as having a neurodegenerative disease-like progression. This category was reserved for those articles that described mild or absent neurological deficits after retirement from boxing, with clear progressive neurological deterioration that had its onset at some later time point and then progressed over a period of years rather than decades. Case 2 reported by Neubuerger et al. is one example, reprinted below:

“His first known admission to a hospital was at the age of 48, for a cholecystectomy. Neurologic and personality changes were not recorded. His wife stated that during the following year he became forgetful, confused, irritable, and moody. He was examined at the Mayo Clinic, where the following observations were recorded: He was an affable, alert, restless patient, disoriented as to time and place, able to perform only the simplest calculations, and unable to find his way about Rochester unescorted. He showed an ataxic gait, decreased speed of motion in the left hand, increased tendon reflexes, and extensor plantar reflexes on the left… The patient’s condition deteriorated progressively. Several hospitalizations were necessary because of confusion, hyperactivity, loquaciousness, and ’nervous breakdown’. He died of progressive pulmonary insufficiency, having required oxygen continuously for the last few months of his life” (page 404) [20].

Interestingly, the autopsy examination of this case showed cerebral atrophy with no neurofibrillary pathology or senile plaques, raising the possibility of frontotemporal lobar degeneration. Another case detailed by Roberts is illustrative, reprinted below:

“Neither he nor his wife had been aware of any neurological disability until he first noticed, at the age of about forty-seven, some difficulty in using his right arm. It was not until five years later that he developed a slowly worsening tremor of his arm which, in the course of a few months, involve all four limbs and was associated with occasional falls, sialorrhea and excessive sweating typical of paralysis agitans. His speech had, since then, become faint and his memory poor” (page 115) [32].

Paralysis agitans refers to Parkinson’s disease. The onset of disease well after retirement and clear progression over the course of years suggests a neurodegenerative process, although the etiology of the process, chronic traumatic brain injury versus sporadic (e.g., Parkinson’s disease), remains an open question. A quote from Roberts, in 1969, relating to this complexity, is reprinted below:

“This is a typical case of paralysis agitans… The speed with which the disease has subsequently progressed is in no way uncharacteristic. It is possible that this syndrome is a late sequel of his boxing career, but in the absence of any atypical features it would seem entirely unjustifiable to suggest that at least one case of unrelated idiopathic paralysis agitans might not be encountered in the present random sample of boxers” (page 115) [32].

## 5. Early Versus Late 20th Century

Neurotrauma exposure appeared to be extreme in the early 20th century and not as extreme by the end of the century. Accounts of boxers with several hundred fights were common in the older literature and were absent from the more recent literature. More extreme forms of traumatic encephalopathy with severe disability, in some cases necessitating lifelong care, appeared in the early 20th-century literature [1,14,17,22,33,34] and were less common in the late 20th-century literature, although some cases with dementia appearing later in life were described in the late 20th century [41,43,45].

A review in 2005 examining boxing history between 1931 and 2002 noted a drop in the average career duration from 19 years to 5 years, and a drop in the average number of bouts from 336 to 13. Relatedly, the idea that punch drunk syndrome (i.e., a severe form of traumatic encephalopathy syndrome) might be a “relic of the past” was discussed in an unauthored editorial in the *Journal of the American Medical Association* in 1969.

“Having thus established on the basis of recent studies that the ex-boxer’s encephalopathy is a distinct entity, not uncommon among professional fighters, can we be sure that the condition is not an anachronism, a relic of the past? After all, the 15 cases reported by Johnson trace their beginnings to the 1930’s when there was hardly any medical supervision of boxing. Only the future will tell whether the ex-boxer’s encephalopathy is on its way to becoming an ex-disease. The present can only bid us to be watchful for florid manifestations of its syndrome and for possible, as yet unproven, minor mental changes following ’acceptable’ boxing blows to the head” (page 2272) [56].

As noted above, boxers in the latter part of the 20th century benefited from improved medical oversight [51,53] and more attention to the rules of engagement (e.g., attention to matching fighters evenly by weight class and skill, more inclination for referees to stop fights, larger boxing gloves, mandatory exclusion times, etc. [32,51]), particularly after World War II. Quotes from the book by Roberts, published in 1969, are shown below:

“…there was much that was different in professional boxing before and after the last war which might be of greater relevance than the changes associated with ageing or any postulated progression of the condition in the high frequency of traumatic encephalopathy among the older boxers (pages 53–54)… Medical supervision of the sport was not a prominent feature of professional boxing before the war as it became soon after and has become, increasingly, since (page 56)… Severely disabling degrees of this syndrome were encountered very infrequently, and not at all in those whose professional careers followed the last war” (page 110) [56].

Roberts noted that, among retired boxers aged 50 and older who boxed for more than ten years, 47% showed evidence of brain damage from boxing, compared to 17% who boxed six to nine years, and 13% who boxed no more than five years [32] (see Figure 2). Among boxers aged 30 to 49 years, the percentages were 25%, 14%, and 1% for those who boxed ten years, six to nine years, and no more than five years, respectively [32]. In terms of numbers of fights, about half of the boxers aged 50 and older with 150 or more professional fights showed evidence of encephalopathy, compared to 19% with 50 to 150 fights (see Figure 2).

We attempted to code the case studies for evidence of boxing participation before or after 1945. However, this degree of precision could not be determined in a number of cases, particularly those in the 1969 Roberts [32] and the 1957 Critchley publications [18], which would have resulted in a number of cases being coded as “unclear”. As an alternative approach, we separated cases based on publication year as 1973 and earlier (or Corsellis et al. and earlier) and 1974 and later (post-Corsellis). This roughly sorted cases into earlier and later 20th century, and all cases were included. Corsellis et al., although published in 1973, specifically stated that their 15 subjects fought between 1900 and 1940, and the Roberts study, although published in 1969, specifically noted that his subjects were all registered boxers with the British Boxing Board of Control between 1929 and 1955 [32]. A few boxers described in articles in the 1950s and 1960s would have fought after World War II, but apart from one report of young boxers with brief histories and few neurological findings [31], no other articles published in 1973 or earlier described boxers whose careers would have taken place after the 1950s.

Publications from 1974 and later described boxers whose careers would have taken place in the latter half of the 20th century, with the possible exceptions of two of the three cases described by Hof and colleagues [40]. Although published in 1992, Hof et al. described boxers with 25-year careers and over 600 fights, which suggest early, or at the latest mid, 20th-century boxers. The third case in Hof et al. was previously reported by Constantinidis and Tissot [28] in 1967, as noted above, so it is included in the earlier 20th century group.

The frequencies with which neurological problems were reported in articles from 1973 and earlier were considerably higher for dysarthria, abnormal gait, memory loss, abnormal reflexes, abnormal facies, ataxia, abnormal eye movements, increased tone/spasticity, tremor, headache, Babinski sign, and bradykinesia (see Table 2 and Figure 3). Some type of progressive disease was noted in 62 out of 169 boxers (37%) in articles published in 1973 and earlier, compared to only 5 out of 74 boxers (7%) in articles published in 1974 and later [χ^2^(1) = 23.09, *p* < 0.001; odds ratio (OR) = 8.00, 95% confidence interval (CI) = 3.06–20.89].

## 6. Discussion

The case histories available in the 20th century provide an understanding of the neurological deficits described in boxers believed to have punch drunk syndrome or dementia pugilistica. These individuals had slurring dysarthria, gait disturbances, and short-term memory deficits. These symptoms were often accompanied by bradykinesia, hyperreflexia (often asymmetric), extensor Babinski signs, impassive or immobile facies, increased tone and spasticity, gait disturbances including ataxia, and abnormal eye movements. In occasional cases, vision loss, cogwheel rigidity, intention tremor, clonus, pathological drooling or dribbling, and frank dementia were described. The marked variability among cases was such that a pathognomonic neurological sign or set of neurological signs (i.e., a cohesive neurological syndrome) was not codified in the 20th century; although, some neurological problems tended to occur together (e.g., dysarthria, gait disturbances, tremor, memory loss) in a way that seemed to characterize chronic and sometimes progressive brain damage in some boxers. There was a notable decrease in the frequency with which neurological signs were reported in the articles published in 1974 and later. This is consistent with the previously noted decrease in neurotrauma exposure over the course of the 20th century [32,51] and a so-called “dose-response” relationship between exposure and outcome.

The clinical features described in many of these boxers were characterized by multifocal and heterogeneous neurological deficits that appear to reflect, as described in one case by Parker, “a medley of scattered and incomplete lesions of the brain” [2]. Psychiatric problems often accompanied the neurological deficits, as enumerated in detail by Iverson et al. [54], but psychiatric problems attributed to boxing, in the absence of neurological deficits, were largely absent from this literature. There is an anecdotal account of hysterical blindness in a boxer that appeared outside the ring [9] and accounts of boxers with catatonic schizophrenia [32,33].

Progressive clinical conditions in 20th-century boxers were both complex and diverse. The presence of stable disease (no progression), which Roberts thought was the case for “most” affected boxers, and the occasional cases with symptomatic improvement contradict a basic feature of neurodegenerative diseases, which is invariable and inexorable progression. Interestingly, progression rated by us as complex was only present in articles published prior to 1974 (early to mid 20th-century boxers). Progression in such cases was apparent often over several decades of life, with significant neurological deficits dating to boxing participation, but the relative contributions of boxing-related neurotrauma, versus co-morbidities such as alcoholism, age-related deterioration, neurovascular disease, trauma sustained outside of boxing, and mental health problems, are difficult to disentangle. The clinical vignettes from Corsellis et al. are the most illustrative of this type of progression, noted in eight of the 15 cases. Details of the neurological examination at the time of retirement from boxing were somewhat limited, but the narratives overall provided a vivid description of the plight of the severely brain-damaged boxer. We have included a summary of those cases in Appendix A.

Also interesting were descriptions of progressive signs limited to motor function. Roberts was noteworthy among the researchers in the 20th century as one who commented on this tendency, particularly with respect to extrapyramidal lesions. In some cases, there was progression over a period of years, followed by stabilization, or ”stationary” disease, adding a layer of complexity to conceptualizing progressive disease in boxers.

Articles that described neurological progression similar to what is seen in canonical neurodegenerative diseases (such as Alzheimer’s disease, frontotemporal dementia, and Parkinson’s disease) were relatively uncommon. The clinical descriptions in such cases often suggested a neurologically intact boxer at the time of retirement and a clear neurological deterioration usually over a period of years. One noteworthy example is the first reported autopsy examination of a boxer, showing smoothly progressive neurological deterioration starting at age 40, and with widespread “condensation plaques” and severe cerebral amyloid angiopathy at autopsy ten years later [14]. The possibility that this boxer had a disease caused by presenilin-1 mutation, rather than boxing-related chronic TBI, has been noted in the recent literature [53,57].

The severity of the condition in boxers, particularly those who fought in the early part of the 20th century, might not be fully appreciated. Current discussions of traumatic encephalopathy syndrome and chronic traumatic encephalopathy often invoke Martland and early 20th-century boxers as the recognition of an entity [58,59], but they do not point out that punch drunk syndrome, or dementia pugilistica, was markedly different from chronic traumatic encephalopathy and traumatic encephalopathy syndrome described today. One could even suggest that the classical condition and the present day concepts are mutually exclusive, insofar as punch drunk syndrome or dementia pugilistica was a clinical/neurological disease state brought to the attention of medical science by lay observers [1], while recently proposed chronic traumatic encephalopathy neuropathology is a postmortem neuropathological entity [60,61], and it is not known whether this neuropathology causes specific neurological or psychiatric problems or whether it is inevitably progressive [59].

Our findings in this review in comparison with the 21st-century literature tend to agree with the suggestion in 1969 that florid manifestations of traumatic encephalopathy were largely confined to the early 20th century [56]. The prototypical boxer of that era described as having severe traumatic encephalopathy was an ultra-high exposure fighter with a lengthy career, disfiguring facial features, and a myriad of abnormalities on neurological examination, including combinations of pyramidal, extra-pyramidal, pseudobulbar, ocular, and neurocognitive dysfunction. Some of these boxers had obvious neurological problems associated with chronic traumatic brain injury while still actively fighting. There were examples of severe cognitive impairment, and even dementia, prior to retirement from boxing.

## 7. Limitations

The major limitation of this review and analysis is the heterogeneity in clinical information provided in the various articles. Some articles provided more detail than others, and even long-term follow up [2], while others were limited to cursory short paragraphs [18,29] or lines in tables [36,43]. Incomplete information across numerous articles could have biased the findings, depending on the characteristics of the missing information. The descriptions of the neurological examinations in particular were lacking in detail, such that it is difficult to exclude disease processes unrelated to trauma in any individual case or examination. For example, conditions related to chronic alcohol abuse (e.g., thiamine deficiency), causes of symmetrical hyperreflexia such as hyperthyroidism, or B12 deficiency (such as in cases with positive Romberg sign) could be suggested in some cases, which could have biased the results if they were not appropriately diagnosed. The description of gait, memory loss, headaches, bradykinesia, abnormal facies, eye movement disorders, abnormal speech (apraxia of speech versus dysarthria), and seizures (focal, generalized), when present, often lacked sufficient detail to completely exclude non-trauma related neurological problems. A small number of sporadic and genetic neurodegenerative diseases were also likely present among the cases, as noted above.

It should also be emphasized that the statistical calculations were carried out with the assumption that neurological signs “not mentioned” were absent, which introduces an error that could impact statistical significance. It seems unlikely that researchers in the latter part of the 20th century would have missed neurological examination findings to any substantial degree. For example, three studies (Kaste et al. [36], Casson et al. [37], and Jordan et al. [43]) comprise 62 of the 74 post-1973 cases. In each of these studies, the authors indicate that all subjects underwent “detailed neurological examination” (Kaste et al., Jordan et al.) or “formal neurological examination including mental status” (Casson et al.), in addition to batteries of neuropsychological tests. Overall, it seems unlikely that the major observation in this review, which is the significant decrease in neurological deficits in boxers commensurate with decreased neurotrauma exposure over the course of the 20th century, would have been altered by potential findings that were coded as “not mentioned”.

## 8. Summary

Neurotrauma associated with boxing was substantial in the 20th century and often resulted in a neurological disorder characterized by multiple neurological deficits, especially slurring dysarthria, gait problems, and cognitive impairment. Other signs such as hyperreflexia (often with left greater than right asymmetry), impassive facies, extensor Babinski sign, increased tone, and ataxia were also commonly reported. Severe examples seemed to be limited to boxers who fought in the early 20th century and could be associated with frank dementia, sometimes requiring lifelong care. Over the course of the 20th century, there was a decrease in individual boxing-related neurotrauma exposure, as measured by number of fights and career duration, which was accompanied by a notable decrease in neurological signs reported in boxers. Progressive disease among 20th-century boxers was variable, and the specific manifestations of progression were diverse. The absence of progression and symptomatic improvement were both noted in some boxers, and neurodegenerative disease-like progression was relatively uncommon.

## 9. Directions for Future Research

A significant challenge in the 21st century is finding the distinction between traumatic encephalopathy syndrome (TES) as recently proposed [50] and known canonical neurodegenerative diseases including Alzheimer’s disease, idiopathic Parkinson’s disease, amyotrophic lateral sclerosis (ALS), and frontotemporal lobar degeneration (FTLD) that could fully account for a given neurological presentation. As noted in this review, sporadic neurodegenerative diseases were present in some boxers, while one case had features suggesting autosomal dominant disease [14,53,57] and another case had a family history of ALS and a *C9orf72* mutation [45,62]. Future research aimed at better excluding sporadic and genetic neurodegenerative diseases might help avoid misdiagnosing individuals with these diseases as having TES and direct patients and their families to genetic counseling when needed.

More research into the distinction between TES and idiopathic Parkinson’s disease is needed. In theory, Parkinson’s disease could fully explain a progressive neurological disease and would therefore exclude TES, yet bradykinesia, rigidity, rest tremor, and Parkinsonian gait disorder are all supportive features of TES [50]. Moreover, it is not clear whether CTE neuropathology, as currently defined [60,61], is associated with the clinical features of Parkinsonism. In 2021, a large-scale clinicopathological study designed to validate the 2014 criteria for TES was published, and in those 336 brain donors, 244 (72.6%) were identified as having CTE neuropathology, and 92 (27.4%) did not have CTE neuropathology [63]. The authors reported that there was not a significant association between having “motor symptoms” and having CTE neuropathology.

In the present review, dysarthria was one of the most common neurological signs in boxers in the 20th century, especially in the ultra-high exposure boxers from earlier in the 20th century. According to the new consensus criteria for TES, published in 2021 [50], dysarthria is one of the motor signs that is considered to be a supportive feature for determining the provisional level of certainty that a person is harboring CTE neuropathology. However, the best available evidence suggests that dysarthria is not associated with CTE neuropathology. In a large-scale clinicopathological study [63], 17.6% of those who had CTE neuropathology had dysarthria compared to 28.3% of those who did not have CTE neuropathology (see online supplementary material eTable 5 of the article by Mez and colleagues [63]). There was actually a statistically significant difference between groups, whereby those who had CTE neuropathology were *less* likely to have dysarthria during life compared to those who did not have CTE neuropathology [χ^2^(1) = 4.63, *p* = 0.031, OR = 0.54, 95% CI = 0.31–0.95].

In the past 15 years, there has been a strong emphasis on psychiatric disorders, substance abuse disorders, and psychosocial problems as being characteristic of CTE and TES by some authors [64,65,66], whereas other authors have reported that these problems do not appear to be reliable, specific, or characteristic diagnostic features of TES [54,55,67,68,69]. It is important to appreciate that, in a previously published large clinicopathological association study, there were no differences between those with CTE neuropathology and those who did not have the neuropathology in depressive symptoms, hopelessness, suicidality, anxiety, social inappropriateness, impulsivity, alcohol use disorder, explosivity, verbal violence, physical violence, apathy, or paranoia (see online supplement of the article by Mez and colleagues [63], eTable 5). There was a statistically significant difference between groups in the proportions who had a substance use disorder that was not related to alcohol or marijuana. Those who had CTE neuropathology were *less* likely to have a substance use disorder during life compared to those who did not have CTE neuropathology [31.6% versus 44.6%; χ^2^(1) = 4.96, *p* = 0.026, OR = 0.57, 95% CI = 0.35–0.94]. The recent consensus statement from the Concussion in Sport Group, published in 2023, noted that it is not known whether CTE neuropathology causes specific neurological or psychiatric problems or whether the neuropathology is inevitably progressive [70]. Future rigorous and transparent clinicopathological studies are needed.

## Figures and Tables

**Figure 1 brainsci-15-00729-f001:**
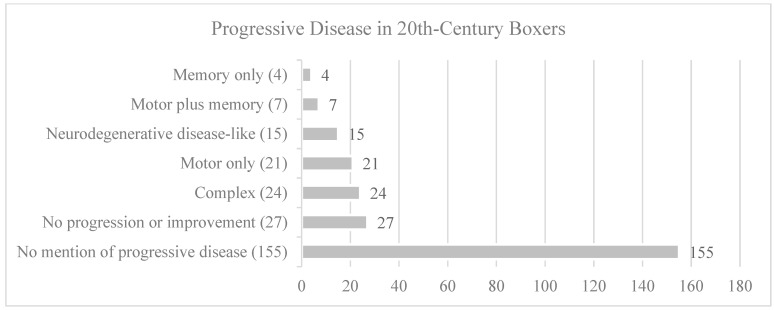
Progressive disease characteristics and frequencies.

**Figure 2 brainsci-15-00729-f002:**
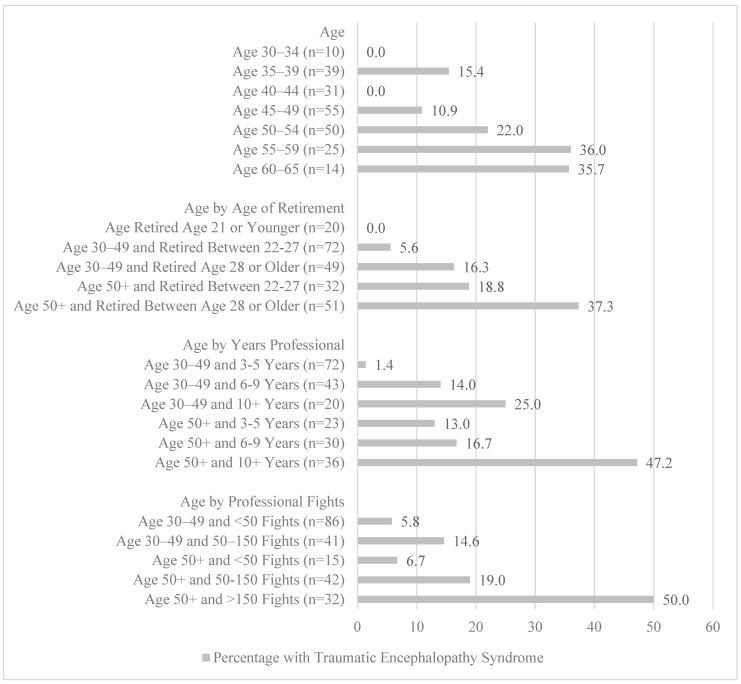
Percentages of former professional boxers with traumatic encephalopathy from a sample of 221 former boxers examined by Roberts and published in a book in 1969. Note: The data presented in this figure were extracted from Table 6 (page 54) and Table 7 (page 58) from [32].

**Figure 3 brainsci-15-00729-f003:**
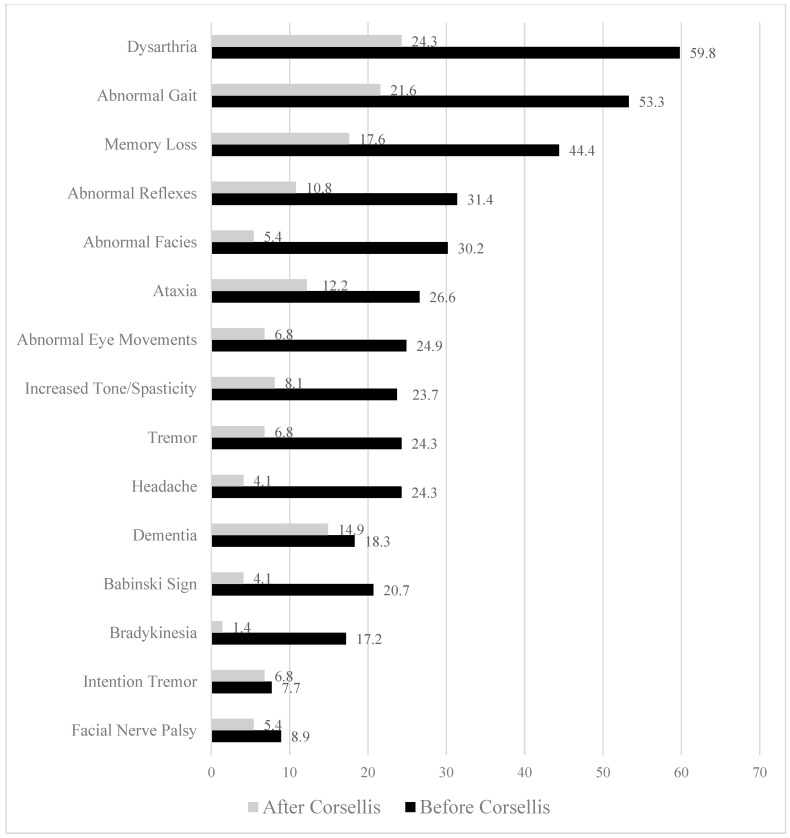
Frequency (percentages) of neurological signs reported in articles published before and after 1974.

**Table 1 brainsci-15-00729-t001:** Articles with cases included in this review.

Article	Cases
Martland (1928) [1]	1
Parker (1934) [2]	3
Herzog (1938) [3] *	9
Knoll (1938) [4] *	3
Bowman and Blau (1940) [5]	1
Grewel (1941) [6] *	1
Bourrat and Micolier (1944) [7] *	2
McAlpine (1949) [8]	1
Critchley (1949) [9]	7
Raevuori-Nallinmaa (1950) [10]	2
Schwarz (1953) [11] *	1
Taylor (1953) [12]	1
Soeder and Arndt (1954) [13] *	5
Brandenburg and Hallervorden (1954) [14] *	1
Schwarz (1955) [15] *	1
Bergleiter and Jokl (1956) [16] *	1
Grahmann and Ule (1957) [17] *	4
Critchley (1957) [18]	17
Muller (1958) [19] *	3
Neubuerger et al. (1959) [20]	2
Wolowska (1960) [21] *	1
Courville (1962) [22]	1
Spillane (1962) [23]	5
Mawdsley and Ferguson (1963) [24]	10
Huszar and Kornyey (1965) [25] *	1
Burger and Marinovjech (1966) [26] *	1
Goralski and Sypniewski (1967) [27] *	1
Constantinidis and Tissot (1967) [28]	1
Payne (1968) [29]	6
Betti and Ottino (1969) [30]	1
Bousseljot (1969) [31] *	8
Roberts (1969) [32]	48
Johnson (1969) [33]	4
Corsellis et al. (1973) [34]	15
Harvey and Davis (1974) [35]	1
Kaste et al. (1982) [36]	14
Casson et al. (1984) [37]	18
Sabharwal et al. (1987) [38]	4
Friedman (1989) [39]	1
Hof et al. (1992) [40]	2
Jordan et al. (1995) [41]	1
Geddes et al. (1996) [42]	1
Jordan et al. (1997) [43]	30
Geddes et al. (1999) [44]	1
(no author listed) (1999) [45]	1

Note: * Data extraction was obtained from 16 articles translated in *Boxing: Medical Aspects* [46]. See Appendix A for the raw data used in this study.

**Table 2 brainsci-15-00729-t002:** Neurological signs or symptoms.

	Total Sample		Before Corsellis (1973)		After Corsellis (1974)	
Sign or Symptom	Present	% Present	Present	% Present	Present	% Present
Dysarthria	119	49.0%	101	59.8%	18	24.3%
Abnormal gait	106	43.6%	90	53.3%	16	21.6%
Memory loss *	88	36.2%	75	44.4%	13	17.6%
Hyperreflexia **	61	25.1%	53	31.4%	8	10.8%
Abnormal facies	55	22.6%	51	30.2%	4	5.4%
Ataxia ***	54	22.2%	45	26.6%	9	12.2%
Eye movement disorder or other ^#^	47	19.3%	42	24.9%	5	6.8%
Increased tone and/or spasticity	46	18.9%	40	23.7%	6	8.1%
Tremor NOS	46	18.9%	41	24.3%	5	6.8%
Headache	44	18.1%	41	24.3%	3	4.1%
“Dementia”	42	17.3%	31	18.3%	11	14.9%
Babinski Sign (plantar extensor reflex)	38	15.6%	35	20.7%	3	4.1%
Bradykinesia	30	12.3%	29	17.2%	1	1.4%
Intention tremor ^##^	18	7.4%	13	7.7%	5	6.8%
Facial nerve paralysis	15	6.2%	15	8.9%	0	0%
Vision loss	14	5.8%	14	8.3%	0	0%
Epilepsy or seizures	14	5.8%	13	7.7%	1	1.4%
Romberg sign	13	5.3%	9	5.3%	4	5.4%
Cogwheel/rigidity	13	5.3%	6	3.6%	7	9.5%
Dysdiadochokinesis	11	4.5%	9	5.3%	2	2.7%
Clonus	11	4.5%	10	5.9%	1	1.4%
Pathological drooling or dribbling	10	4.1%	9	5.3%	1	1.4%

Note: NOS = not otherwise specified. * Memory loss includes subjective (subject or family report) and objective (examiner-determined). ** Reflexes were coded separately as increased not otherwise specified, increased left > right, increased right > left, decreased. *** Ataxia was coded both as ataxia and as abnormal gait, unless referring to another modality (e.g., ataxic speech). ^#^ Any abnormality related to ocular function, other than vision loss, was coded as eye movement disorder or other. ^##^ “Intention tremor” was coded as tremor and separately as intention tremor.

**Table 3 brainsci-15-00729-t003:** Frequency of disease progression ratings among cases in which progression is discussed.

Progression Rating	*n*	*n* = 88 % *	*n* = 243 %
Progressive condition (any)	**71**	**81%**	**29%**
Complex deterioration	24	27%	10%
Progression of motor deficits	21	24%	9%
Canonical neurodegenerative disease-like progression	15	17%	6%
Progression limited to motor deficits and memory loss	7	8%	3%
Progression of memory loss only	4	5%	2%
No progression/stationary condition or symptomatic improvement	27	31%	11%
No progression/stationary	19	22%	8%
Symptomatic improvement	8	9%	3%

Note: * Because some cases had more than one rating, the percentages added up to more than 100%. There was a subgroup of 88 cases in which progression was mentioned. The total sample was 243. Complex deterioration was over decades with medical co-morbidities such as alcoholism, vascular disease, and traumatic brain injury sustained outside of boxing. Canonical neurodegenerative disease-like progression was progression over a period of years beginning after retirement, with apparently preserved neurological function at baseline (i.e., progression similar to known neurodegenerative diseases).

## Data Availability

The original contributions presented in the study are included in the article and in Appendix A.

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
