# Peer review of "Neurological Disorders and Clinical Progression in Boxers from the 20th Century: A Narrative Review†"

_brainsci, 2025, doi:10.3390/brainsci15070729_

Round 1

Reviewer 1 Report

Comments and Suggestions for Authors

The review titled Neurological Disorders and Clinical Progression in Boxers 2 from the 20th Century: A Narrative Review provides valuable information on the neurological signs documented in boxers throughout the 20th century, with particular attention to the evolution of neurological damage associated with repeated trauma exposure in boxing. However, I have several comments that should be considered:

It is necessary to be more explicit about the rationale for reviewing this specific time period.

The manuscript covers a broad period (the entire 20th century), which offers an ideal opportunity for a comparative analysis across different timeframes for example, the early decades of the 20th century (1900–1949) versus the later decades (1950–1999), or before and after the implementation of boxing safety measures.

Although the number of reviewed cases is mentioned, it is unclear whether a systematic review methodology was applied, and what the search, inclusion, and exclusion criteria for studies and cases were.

It is noted that two papers were translated into English; however, the manuscript does not specify in which languages the literature search was conducted. If searches were performed in other languages, why were only two non-English studies included?

Regarding the included cases, it would be worthwhile to mention whether they involved amateur or professional boxers. If information on both groups was available, a comparative analysis of neurological disorders in amateur versus professional boxers would be of great interest.

The authors should discuss how the heterogeneity of reports may have biased the clinical categorization or the conclusions regarding disease progression.

The authors mention that some neurological signs observed in boxers could be explained by other conditions unrelated to repeated head trauma, such as vitamin B12 deficiency, hyperthyroidism, or alcohol abuse. However, they do not elaborate on the extent to which or how these alternative diagnoses may have biased the results, nor whether such cases were excluded or how they were managed.

Author Response

See attached reply to reviewer 1 and 2. 

Reviewer 2 Report

Comments and Suggestions for Authors

I appreciate the invitation to review the manuscript titled “Neurological Disorders and Clinical Progression in Boxers From The 20th Century: A Narrative Review” submitted for publication in Brain Sciences.

In the present study,researchers studied the history of neurological issues in 20th-century boxers to better understand chronic traumatic encephalopathy (CTE) and traumatic encephalopathy syndrome (TES), which currently lack clear diagnostic criteria. Analyzing 45 articles from 1928 to 1999 that described 243 boxers, they found common symptoms like slurred speech, gait problems, and memory loss. Boxers from the latter half of the century had fewer neurological issues. The progression of symptoms was often unclear, but some cases showed progressive or stable conditions. These historical findings differ from modern TES criteria, highlighting the complexity of diagnosing these conditions.

Overall, I believe that the work is innovative, robustly conducted, comprehensive and of high interest/relevance. I commend the authors for their efforts. I have the following comments for them to consider and refine their manuscript.

  1. The introduction part needs to be re-written and emphasis should be on why such a review is crucial. In the introduction part historical context, advancement in diagnostic criteria, how reviewing past case studies can help researchers better understand the progression of neurological damage and identify risk factors, psychiatric symptoms, should be discussed rather than discussing the description of past literature.
  2. Table 2: Neurological signs and symptoms: The table presenting neurological signs and symptoms in Table 2 appears to be identical to one previously published by the authors. To avoid duplication, it would be preferable for the authors to either reformat the table or provide a reference to the original publication where the table first appeared.

Author Response

Please see attached reply to reviewer 1 and 2. 

Round 2

Reviewer 1 Report

Comments and Suggestions for Authors

I have reviewed the authors’ responses and the revised version of the manuscript entitled “Neurological Disorders and Clinical Progression in Boxers from the 20th Century: A Narrative Review.” I commend the authors for thoroughly addressing the previous comments and improving the manuscript significantly.

The manuscript has been substantially strengthened and now meets the standards for publication.